# Comparison of the Morphological and Structural Characteristic of Bioresorbable and Biocompatible Hydroxyapatite-Loaded Biopolymer Composites

**DOI:** 10.3390/nano11123194

**Published:** 2021-11-25

**Authors:** Monika Furko, Zsolt E. Horváth, Judith Mihály, Katalin Balázsi, Csaba Balázsi

**Affiliations:** 1Centre for Energy Research, Institute of Technical Physics and Materials Science, Konkoly-Thege Str. 29-33, 1121 Budapest, Hungary; horvath.zsolt.endre@ek-cer.hu (Z.E.H.); balazsi.katalin@ek-cer.hu (K.B.); csaba.balazsi@ek-cer.hu (C.B.); 2Research Centre for Natural Sciences, Institute of Materials and Environmental Chemistry, Magyar Tudósok Körútja 2, 1117 Budapest, Hungary; mihaly.judith@ttk.mta.hu

**Keywords:** carbonated hydroxyapatite, electrospinning, biopolymers, morphology, biodegradable

## Abstract

Calcium phosphate (CaP)-based ceramic–biopolymer composites can be regarded as innovative bioresorbable coatings for load-bearing implants that can promote the osseointegration process. The carbonated hydroxyapatite (cHAp) phase is the most suitable CaP form, since it has the highest similarity to the mineral phase in human bones. In this paper, we investigated the effect of wet chemical preparation parameters on the formation of different CaP phases and compared their morphological and structural characteristics. The results revealed that the shape and crystallinity of CaP particles were strongly dependent on the post-treatment methods, such as heat or alkaline treatment of as-precipitated powders. In the next step, the optimised cHAp particles have been embedded into two types of biopolymers, such as polyvinyl pyrrolidone (PVP) and cellulose acetate (CA). The pure polymer fibres and the cHAp–biopolymer composites were produced using a novel electrospinning technique. The SEM images showed the differences between the morphology and network of CA and PVP fibres as well as proved the successful attachment of cHAp particles. In both cases, the fibres were partially covered with cHAp clusters. The SEM measurements on samples after one week of immersion in PBS solution evidenced the biodegradability of the cHAp–biopolymer composites.

## 1. Introduction

It is crucial to ensure the long-term success of implanted materials in all orthopaedic surgeries. To obtain this, a high level of biocompatibility and bioactivity are required from the implants. Generally, the materials used in bone repair are made of metals with high mechanical strength and ductility [1,2]. The metallic implant materials that are implanted in the human body should withstand aggressive environment with several ions present in the body fluid with pH 7 and temperature of 37 °C [3]. Thus, the high corrosion resistance of these materials is also extremely important. To date, there has been intensive research to increase the biocompatible characteristics of implants. This can be best achieved by applying appropriate calcium phosphate-based coatings on their surface. The properties and biological performance of coatings can be adjusted to meet the standard requirements of biomedical applications [4,5]. The main approach is that the bioceramic coatings act as an initiative intermediate surface for the increase in bone cell attachment, growth, and proliferation as well as new bone formation [6,7,8,9]. Once the bones are repaired, the in vivo degradation of coatings is favourable from both clinical and biomedical points of view. The biodegradability of bioceramic coatings can be modified by incorporating them into adequately chosen biopolymer fibres. Suitable biopolymers can be the cellulose acetate (CA) and the poly N-vinyl pyrrolidone, (PVP). It is also an important factor that the bioceramic–biopolymer composite coatings degrade to non-toxic by-products, being removed by the body completely after ossification [10,11]. PVP is a synthetic polymer with unique properties of biodegradability in a biological environment [12] and also an amphiphilic molecule owing to the polar lactam group in pyrrolidone offering hydrophilicity and non-polar methylene part providing lipophilic characteristics [13]. The PVP gained more attention in the field of biomedical application because of its inertness, chemical stability, non-toxicity, and biocompatibility. There has also been reported research on the composite formation of PVP with other biomaterials such as bioceramics and pharmaceutics [14,15,16]. On the other hand, cellulose is a natural, biodegradable, and biocompatible polymer. Cellulose acetate (CA) is the acetate ester of cellulose produced from cellulose through acetylation [17]. CA has unique properties, which make it ideal for many applications, such as filtration, medical coatings, drug delivery, and food packaging [18]. Moreover, CA also has good fibre-forming ability in the electrospinning process. In fibre form, it possesses a high surface area to volume ratio, flexibility, and high porosity [19]. Electrospinning is an easy and economic method to prepare nanofibers through the application of a high electrical field [20]. Generally, when the applied electrostatic voltage exceeds the surface tension of a polymer drop, a charged polymer jet is formed and pumped through a needle. Owing to the high potential difference between the needle and collector, the polymer jet will stretch and elongate and subsequently collect on the grounded collector [19]. The electrospinning process enables easy incorporation of different nanoparticles into the fibre mats by dispersing them into the base polymer solution.

The incorporation of HAp nanoparticles also has an influence on the morphology and structure of electrospun fibres. It was observed that the fibres tended to become thicker and more entangled by increasing the amount of HAp particles [21]. However, an appropriate regulation of the electrospinning parameters can result in relatively homogeneous distributions of inorganic particles without the formation of HAp agglomerates between the fibres [16,22]. In our work, we prepared different calcium phosphate phases by chemical precipitation and optimised the post-treatment methods to obtain the optimal carbonated HAp phase. In addition, we incorporated the cHAp powder into a natural and a synthetic polymer by electrospinning. We make a thorough study and compare the morphological changes caused by the powder addition to the structure of base polymer fibre mats, which has not yet reported in the scientific literature in detail. Immersion tests have also been done to check the changes in the microstructure of composite samples.

## 2. Materials and Methods

### 2.1. Preparation of Different Calcium Phosphate Nano-Powders

Different calcium phosphate phases were prepared by the wet precipitation method, dissolving calcium acetate (Ca(CH_3_COO)_2_·H_2_O, ≥99.0%. VWR International Ltd.—Radnor, PA, USA) and disodium hydrogen phosphate (Na_2_HPO_4_, VWR International Ltd.—99%, AnalaR NORMAPUR, Radnor, PA, USA) in distilled water in a 5:3 mole ratio using a magnetic stirrer (1400 rpm) at room temperature. Then, the as-prepared white precipitations were subjected to either alkaline treatment with 50 g/L sodium carbonate anhydrous, (Na_2_CO_3_, ≥99.5% ACS, VWR International Ltd.—Radnor, PA, USA) at pH value of 11. for 24 h, or heat treatment at 650 °C and 900 °C for 5 h. The powders were collected and used for further characterisation and for the electrospinning process.

### 2.2. Preparation of cHAp/PVP and cHAp/CA Composites by Electrospinning

PVP (polyvinylpyrrolidone, average M_w_ ≈ 1,300,000, Sigma-Aldrich, St. Louis, MO, USA) and cellulose acetate (average M_w_ ≈ 100,000, Acros Organics, Geel, Antwerp, Belgium, acetyl content 39.8%) were used as biopolymer matrices. The electrospinning was performed using high-voltage supply with an Inovenso Ne100 Electrospinning/Electrospraying Machine (maximum 35 kV, Inovenso Inc., Boston, MA, USA). The solvents were 96% ethanol and acetone for PVP and CA electrospinning, respectively. The electrospun fibres were collected on a parallel collector wrapped with aluminium foil, which was placed at a distance of 12 cm away from the needle tip. A 10 mL plastic syringe was filled with the prepared suspensions and was pumped through a metal needle with a diameter of 0.95 mm with a flow rate of 1 mL/h, and a high voltage of 20 kV was applied between the two electrodes. The process was performed at room temperature. In order to form fibrous biopolymer composite fibres loaded with cHAp particles, 10 wt% PVP/1 wt% cHAp suspension was made in 96% ethanol media as well as 10 wt% CA/1 wt% cHAp was dissolved in acetone. The flow rate in these cases was increased to 1.5 mL/h due to the increased viscosity of the suspensions. All the other parameters (electrode distance, voltage) remained the same. A schematic presentation of preparation methods can be seen in Figure 1.

### 2.3. Characterisation Methods

#### 2.3.1. X-Ray Diffraction Analysis

Calcium phosphate phases were characterised by X-ray diffractometry (XRD, Bruker AXS D8 Discover with Cu Kα radiation source, λ = 0.154 nm) equipped with a Göbel mirror and scintillation detector (Bruker AXS, Karlsruhe, Germany). The equipment was operated at 40 kV and 40 mA. The diffraction patterns were collected over a 2θ range from 10° to 65° with an 0.3°/min steps and 0.02° step size. Diffrac.Eva software was used to evaluate the measured XRD patterns and to identify the crystallite phases.

#### 2.3.2. Scanning Electron Microscopy (SEM)

The morphological properties of calcium phosphate powders as well as the pure and bioceramic-loaded PVP and CA fibres were examined by field-emission scanning electron microscope (SEM, Thermo Scientific, Scios2, Waltham, MA, USA) and Energy-Dispersive X-ray Spectrometry (Oxford Instrument EDS detector X-Max^n^, Abingdon, UK). The map sum spectra were recorded on samples using 6 keV accelerating voltage.

#### 2.3.3. FT-IR and Raman Analysis

Infrared spectroscopic investigation was performed using the attenuated total reflection (ATR) technique. A Varian Scimitar 2000 FT-IR spectrometer (Varian Inc., Palo Alto, CA, USA) equipped with an MCT (mercury–cadmium–telluride) detector was used and fitted with a ‘Golden Gate’ single reflection diamond ATR unit (Specac Ltd., Cray Ave, Orpington, UK). During the measurements, the solid samples, without any sample preparation, were pressed with a constant 70 cNm pressure on the top of the diamond ATR crystal by a sapphire anvil. All spectra were collected with a nominal resolution of 4 cm^−1^ by the co-addition of 128 individual spectra. Before spectral evaluation, ATR correction was performed. Raman spectra were recorded with a Bio-Rad (Digilab) dedicated FT-Raman spectrometer equipped with a Spectra-Physics Nd-YAG-laser (1064 nm) and liquid-N_2_ cooled Ge detector. The power of the excitation laser was about 500 mW at the samples. All spectra were collected using a backscattered geometry with a nominal resolution of 4 cm^−1^ and by the co-addition of 128 individual scans.

#### 2.3.4. Immersion Tests

The bioresorbable characteristics of HAp–biopolymer samples were studied by immersion tests. All the samples were soaked in commercial phosphate buffer saline (PBS, VWR International Ltd. Radnor, PA, USA). The pH of the stock solution was 7.4 with a chemical composition of 137 mM NaCl, 2.7 mM KCl, 8 mM Na_2_HPO_4_, and 2 mM KH_2_PO_4_. During the soaking procedure, the composite samples deposited on aluminium substrate were immersed in 10 mL PBS solution in separate containers at room temperature for one week. The aim of the immersion test in this study was to check the qualitative change in the morphology of polymer and ceramic particles after immersion without any quantitative analysis.

## 3. Results and Discussion

### 3.1. Morphological and Structural Characterisation of Calcium Phosphate (CaP) Powders Prepared with Different Parameters

#### 3.1.1. Scanning Electron Microscope Analysis of CaP Powders

The morphology of calcium phosphate particles prepared by wet chemical precipitation and different post-treatment were investigated and compared.

Figure 2 clearly demonstrate the difference in morphology of CaP powder prepared and treated in different ways. The as-prepared powder using Ca acetate as precursor has particles in the shape of large, thin plates in several micrometre sizes. These plates tend to be oriented parallelly (Figure 2a). This type of morphology is characteristic of the monetite phase, as it is widely discussed in the scientific literature [23,24,25]. After heat treatment, the morphology of particles changed noticeably. Keeping the sample at 650 °C for 5 h resulted in long, rod-shaped particles and smaller cubes attached together as well as large agglomerated blocks (Figure 2b). This change in morphology can be attributed to the phase transformation to calcium pyrophosphate. This is in agreement with other research works where the morphology of metastable Ca pyrophosphate phase was studied [26,27,28]. Annealing the CaP1 sample at 900 °C for 5 h also caused different morphology. In this case, the sample consisted of small, globular-like and rod-like particles in nanometre size. The most noteworthy change is the drastic decrease in the size of the individual particles (Figure 2c). The size of particles ranges between 50 and 300 nm compared to the several micrometre-sized (≈1–4 µm) particles generated at lower temperatures. The alkali treatment of CaP1 sample also resulted in very small needle or thorn-like particles in nanometre size (around 100–150 nm in length). These particles are relatively well oriented and aligned in parallel to each other (Figure 2d). The needle-like shape of each particle is the characteristic form of the hydroxyapatite phase that is described in many scientific works [29,30]. The EDS spectra of different CaP powders were recorded at a large demonstrative, comprehensive area on all samples and are presented in Figure 2e. The calculated atomic percentages are in Table 1. However, it has to be noted that the obtained values from the EDS measurements are only semi-quantitative results, and the data are provided for informative purposes only owing to the low reliability of measurements.

#### 3.1.2. Structural Investigation of Calcium Phosphate Powders Prepared by Wet Precipitation by XRD Measurements

XRD measurements were carried out to determine the phase compositions of powders prepared by different routes (Figure 3). It is visible that the as-prepared powder (CaP1) exhibits only the characteristic peaks of brushite (CaHPO_4_·2H_2_O, JCPDS 01-074-6549), and all the peaks are well-defined and narrow with high intensity, which shows that it is well crystalline. The sample CaP2, on the other hand, has the characteristic pattern of Ca pyrophosphate (Ca_2_P_2_O_7_, JCPDS 01-081-2257). The intensities of peaks are also high and the peaks are narrow and well defined, proving the high crystallinity of powder. The diffractogram of the HAp1 sample reflects the characteristic hydroxyapatite pattern with the strongest triplet of hydroxyapatite crystal at 2*θ* = 31.7°, 32.2°, and 32.9° and indexed as 211, 112, and 300, respectively (JCPDS 01-086-1199). In this case, the powder is also highly crystalline, and the peaks are narrow and well-distinguished. On the contrary, the XRD pattern of the HAp2 powder has broad and merged peaks; however, it also exhibits all the characteristic apatite patterns. The peak broadening, compared to the HAp1 sample, indicates its quasi-amorphous characteristics or the presence of very small, disordered, nano-sized particles [31]. In addition, it is also reported that the anionic HPO_4_^2−^, H_2_PO_4_^−^, CO_3_^2−^, and HCO_3_^−^ contaminants present in the HAp powder could also cause line broadening [32,33]. According to these findings, we also observed notable peak broadening when the as-precipitated CaP powder was post-treated in Na_2_CO_3_ alkaline solution. The broadened peaks in sample HAp2 can also be attributed to the carbonate anion incorporation into the CaP lattice. The diffractograms clearly confirm the CaP phase transformation owing to the heat treatment and alkaline treatment. There are several research works in which the authors have proven that the heat treatment yielded Ca pyrophosphate as an intermediate phase [27,34]. According to the reports and experimental results [29,35,36], the brushite phase starts to lose water at around 196 °C (bound water molecule in crystals) and then transforms to monetite phase. With the annealing temperature increasing, the monetite phase transformed to β-dicalcium pyrophosphate (β-Ca_2_P_2_O_7_) in single phase; then, further increasing the temperature, it was altered to (α-CPPA: α-Ca_2_P_2_O_7_) with lower crystallinity. The second decomposition was attributed to the transformation of the pyrophosphate phase above 400 °C when the monetite loses an H_2_O molecule from two HPO_4_^2−^ groups under high-temperature conditions. They revealed that the improvement of crystallinity depended on the ratio of the HAp phase in the structure, which was managed by the pH value during preparation.

The XRD data of all samples were also analysed by the so-called full pattern fitting [37] of the diffractograms, together with crystallite size calculations, which were executed using the built-in routine of the Diffrac.EVA program. The calculated crystallite sizes (by Scherrer) of powders were 67.3 nm, 57.9 nm, 51.5 nm, and 8.77 nm for CaP1, CaP2, Hap1, and HAp2 samples, respectively.

#### 3.1.3. FT-IR and Raman Characterisation of CaP Powders

Figure 4 illustrates the characteristic FT-IR spectra of different CaP powders.

In well accordance with the XRD results, it is visible that the CaP1 sample is brushite, exhibiting the characteristic peaks of this phase. The bending mode of O–H was recorded at 1647 cm^−1^ [38]. The P-O stretching triplet is present at 1120, 1051, and 980 cm^−1^ [39]. The bands around 570 cm^−1^ can be attributed to the different bending vibrational modes of the P–O bond (O-P-O(H) bending) [29,40].

For the CaP2 sample, the dense frequency band group between 1211 and 1138 cm^−1^ is related to the symmetric P-O and P-O(H) stretching of PO_3_^2–^ groups. The band group between around 400 and 600 cm^−1^ can be ascribed to the asymmetric O-P-O stretching of the PO_4_^3−^ group. These results are in agreement with those of Corrêa et al. [27]. A sharp band is also present at around 721 cm^−1^ owing to P_2_O_7_^4−^ showing the presence of calcium pyrophosphate (Ca_2_P_2_O_7_) [41,42].

The spectra of HAp1 and HAp2 clearly confirm the powders to be carbonated hydroxyapatite. The peaks at around 1401 and 1471 cm^−1^ (ν3) come from the stretching vibrations of CO_3_^2−^ ions and the small peak at 870 cm^−1^ (ν2) shows that the CO_3_^2−^ groups were substituted for PO_4_^3−^ groups, forming B-type carbonated apatite. The spectra of both samples also show the characteristic bands of the PO_4_^3−^ ions ν3 around 1012 cm^−1^ as well as the characteristic splitting of a P-O antisymmetric bond (ν4) at 558 and 599 cm^−1^. The spectra of CaP1, HAp1, and HAp2 show large band between 3500 and 300 cm^−1^, which is due to the O–H stretching vibrations of -OH groups in the bonded water molecule. However, this large band is absent in the case of the CaP2 sample as an intermediate calcium pyrophosphate phase. All the appearing characteristic bands of carbonated HAp are in well accordance with other literature data [43,44,45].

Figure 5 demonstrates the Raman measurements on different CaP powders. The Raman spectra of CaP1 shows the most intense band of ν_1_PO_4_ vibration, which is shifted towards a higher wavenumber (986 cm^−1^), and smaller bands appear at 1130, 1082, 1089, and 880 cm^−1^. These bands are typical of brushite (CaHPO_4_·2H_2_O) [46]. For the CaP2 sample, the IR spectrum indicated Ca pyrophosphate phase, and its Raman spectrum is quite different from those of hydroxyapatites. It is dominated by two bands, namely ν_s_PO_3_ at 1046 cm^−1^ and ν_s_P-O-P at 737 cm^−1^ of the P_2_O_7_^4−^ pyrophosphate ion [47]. On the other hand, both Hap1 and Hap2 samples show typical spectra of carbonated apatite. The strongest Raman band around 959 cm^−1^ belongs to the ν_1_PO_4_ vibration of the apatite structure [48]. The medium band at 1072 cm^−1^ can be assigned to both type-B carbonate (CO_3_ substituting for PO_4_) and ν_3_PO_4_ vibrations [49,50,51]. Regarding the IR spectra, the presence of type-B carbonate is confirmed. It seems plausible that after heat treatment up to 900 °C, free carbonate ions are also present in the sample.

### 3.2. Morphological Characterisation of PURE and cHAp-Loaded PVP and CA

According to the morphological and structural characterisation results of the calcium phosphate phases prepared by different routes, it was revealed that the HAp2 powder has the structure that most resembles that of the mineral phase in natural bones (taking into account the size of the particles, the rate of crystallinity, and the presence of the carbonated hydroxyapatite phase) [52,53,54]. Thus, in the following experiments, we incorporated the HAp2 particles into two different type of biopolymers, namely cellulose acetate and polyvinylpyrrolidone (K85-95).

Figure 6 demonstrates the fibre structures of pure CA and PVP biopolymers. It is clearly visible that the diameters of pure CA fibres vary over a very wide range (from ≈50 nm to 1 µm) with many junctions and some beads within the polymer network. The polymer fibres form wide, long trunks as well as narrow branches (Figure 6a). On the other hand, the diameters of pure PVP fibres show much more homogeneous distribution (between 50 and 200 nm) with entangled, long thread-like fibres, without any beads (Figure 6b). Angel et al. [19] used response surface methodology to study the effect of processing parameters on the electrospun CA. According to their report, the mean diameter of CA fibres ranged from 404 to 1346 nm, and they found that the initial polymer solvent concentration had a profound effect on the resulting fibre diameters (the diameter increased with increasing CA concentration in acetone), while the applied voltage differences and the spinning distance did not show any consistent effect. On the other hand, Lee et al. [55] used dimethylformamide (DMF)/acetone and dichloromethane (DCM)/acetone mixtures as solvents with different volume ratio and studied the relationship between the polymer concentrations as well as the solvent systems in electrospinning. The results revealed that the solvent ratio significantly affected the critical concentration of CA for electrospinning. They obtained porous CA nanofibers from DCM/acetone mixture with the volume ratio of 3:1 and found that low solvent quality worsened the diffusion of polymer filaments, resulting in a porous structure. Similarly, there are an abundant number of research works on preparing PVP fibres [13,16,56,57,58,59], and the reported results are in well agreement with our experiments.

It is visible in Figure 7 that the particles of the HAp2 sample are attached to the CA fibres as clusters, and the cover is discontinuous. In other words, the ceramic particles are incorporated into a mat of interwoven fibres. However, it can also be observed that the huge difference between the diameters of individual fibres, compared to pure CA fibres, lowered (between 100 and 400 nm), which can be attributed to the dispersed cHAp particles and the increased viscosity of suspension.

The cHAp-loaded PVP exhibits noticeably different morphology compared to the cHAp–CA composite (Figure 8). In this case, relatively well aligned, long, thin fibres are visible. The spherical, agglomerated cHAp particles are entwined by the PVP fibres, forming a web-like structure. The size of the cHAp clusters varies over a wide range (1–15 µm).

There are many attempts to prepare hydroxyapatite–biopolymer composites with appropriate bioactive properties for biomedical use or for bone tissue engineering; however, the majority of them use poly(l-lactic acid) (PLLA), poly(e-caprolactone), chitosan, and gelatines matrices [60,61,62,63,64,65,66,67], and only a few of them apply PVP or CA [68,69,70].

### 3.3. Immersion Test

An immersion test was performed to check the biodegradability capacity of composite samples and to get an insight into the morphological change caused by the soaking procedure in general.

Figure 9 demonstrates the microstructure of degraded cHAp–biopolymer composites. The difference in morphology before and after immersion in PBS solution is remarkable for both CA and PVP matrices.

The early stage of dissolution of both polymer fibres is clearly visible. The contours of fibres in these cases are not well-defined and they form a melted structure, encompassing the cHAp particles that have much lower solubility. It can also be observed that the agglomerated larger cHAp clusters, which were found in the case of electrospun cHAp-biopolymer composites (Figure 7 and Figure 8), are broken down into smaller particles with homogeneous distribution on the surface of substrate material (Figure 9b).

In the scientific literature, several reports can be found on the investigation of the dissolution/swelling properties of both CA and PVP polymers and their combination with different materials, using acidic or basic media [71], or even SBF [72]. There are research works also focusing on the quantitative analysis and mechanical tests of samples after immersion [73,74,75,76] and their corrosion behaviour [77,78].

## 4. Conclusions

Different calcium phosphate phases, such as brushit, calcium pyrophosphate, and carbonated HAp powders were successfully prepared by wet chemical precipitation. The precipitation parameters were optimised to obtain a pure carbonated HAp phase that is most similar to the nanostructure and crystallinity of the mineral phase in bones.

Based on the XRD measurements, the heat treatment caused phase transformation to form pyrophosphate and hydroxyapatite phases as the annealing temperature increased and also increased the crystallinity of powders. The alkaline treatment resulted in an almost amorphous phase with weak crystallinity.

The FTIR and Raman spectra also confirmed the presence of all the characteristic peaks of chemical bonds present in calcium phosphates and proved the partial carbonate anionic substitution for PO_4_^2−^ groups.

The successful incorporation of optimised cHAp powder into natural (CA) and synthetic (PVP) biopolymer fibre networks was carried out by a novel electrospinning technique. The SEM images revealed the difference in fibre morphology of pure CA and PVP polymers.

The cHAp powder addition to the base polymer solution also caused a drastic change in the polymer network structure. It was also observed that the carbonated HAp particles were attached onto both types polymer fibres; however, the surface coverage of fibres was discontinuous.

SEM images on samples after the immersion test proven the biodegradability of both types of cHAp–biopolymer composites.

## Figures and Tables

**Figure 1 nanomaterials-11-03194-f001:**
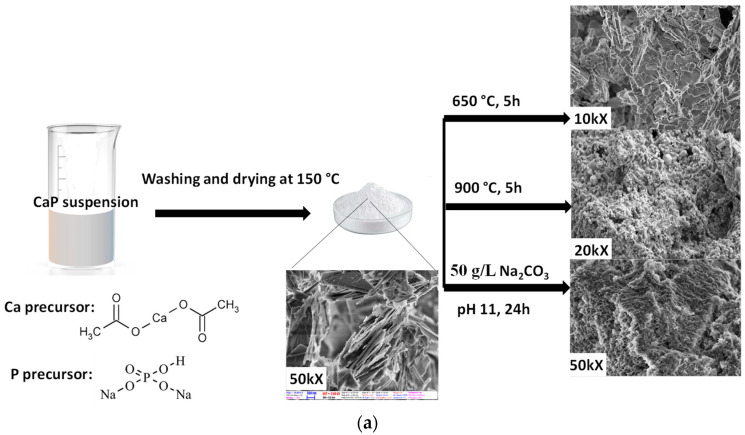
Schematic illustration of the preparation methods (**a**) CaP powder preparation by wet chemical method and (**b**) electrospinning of cHAp–biopolymer composites.

**Figure 2 nanomaterials-11-03194-f002:**
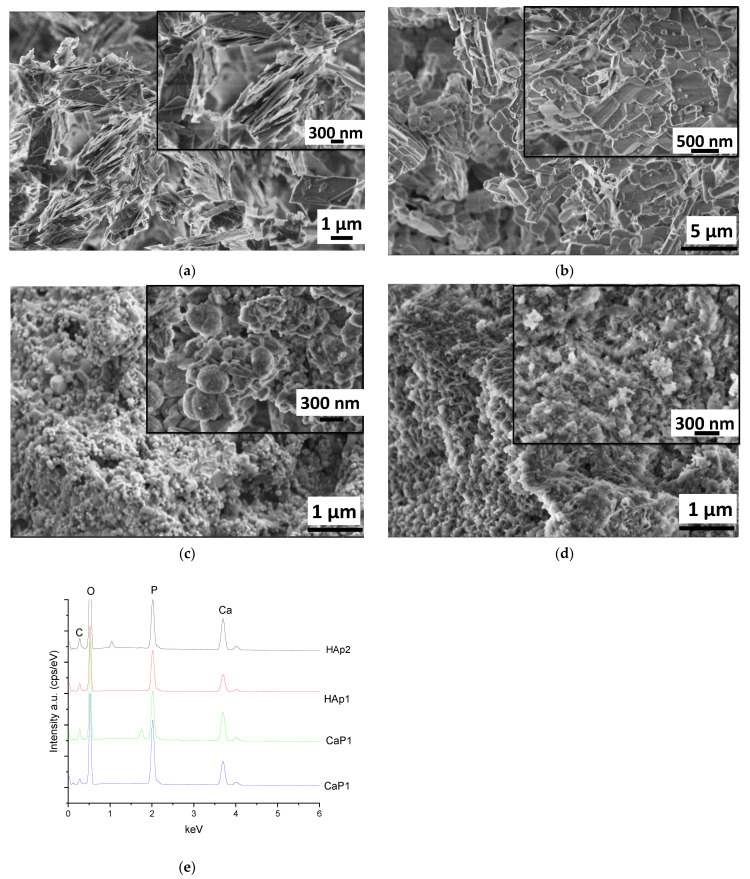
SEM images of calcium phosphate particles (**a**) as-prepared by wet chemical precipitation (sample CaP1): magnification 20 kX, inset 50 kX (**b**) CaP1 sample heat treated at 650 °C for 5 h (sample CaP2): magnification 10 kX, inset 20 kX (**c**) CaP1 sample heat treated at 900 °C for 5 h (sample HAp1): magnification 20 kX, inset 100 kX and (**d**) CaP1 sample alkaline-treated in Na_2_CO_3_ solution for 24 h (sample HAP2): magnification 50kX, inset 100 kX as well as (**e**) an averaged EDS spectra of different samples.

**Figure 3 nanomaterials-11-03194-f003:**
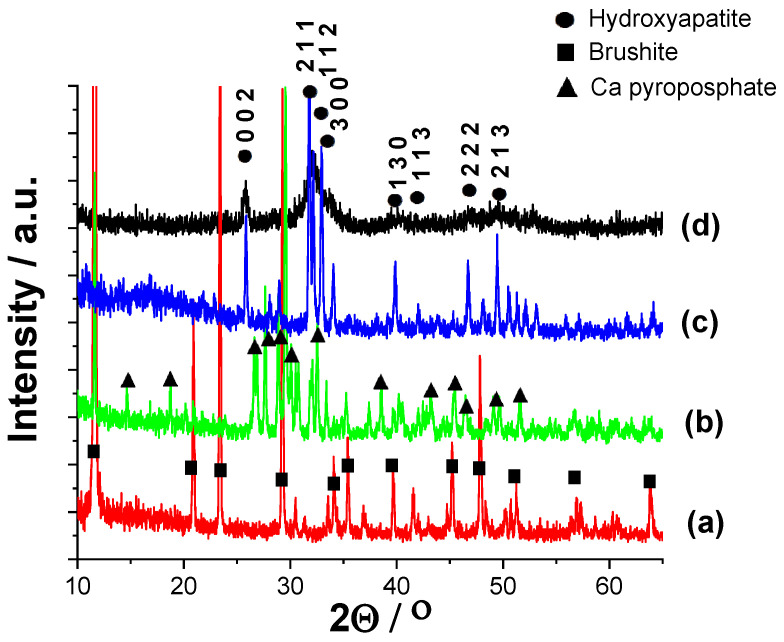
XRD patterns of CaP powders prepared by different parameters (**a**) as prepared by the wet chemical method: CaP1 (**b**) CaP1 sample heat treated at 650 °C: CaP2 (**c**) CaP1 sample heat treated at 900 °C: HAp1 and (**d**) alkaline treatment in Na_2_CO_3_ solution at pH 11 for 24 h: HAp2.

**Figure 4 nanomaterials-11-03194-f004:**
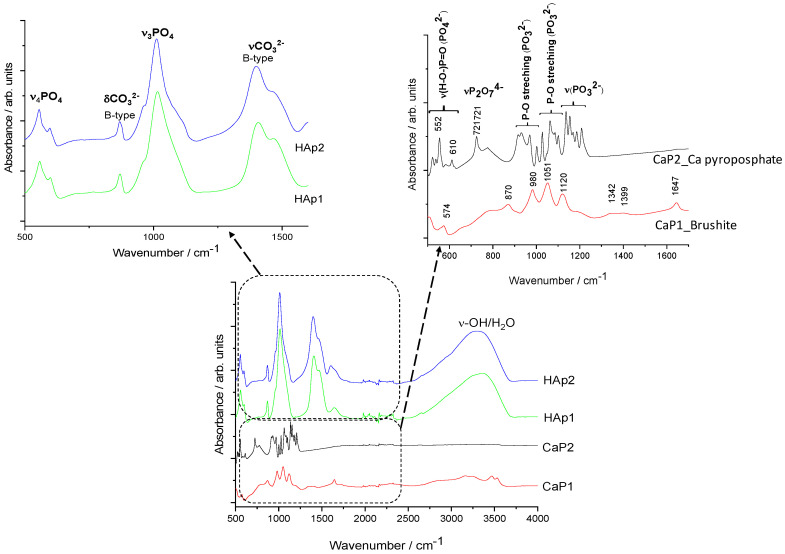
FT-IR spectra of calcium phosphate powders prepared by different routes.

**Figure 5 nanomaterials-11-03194-f005:**
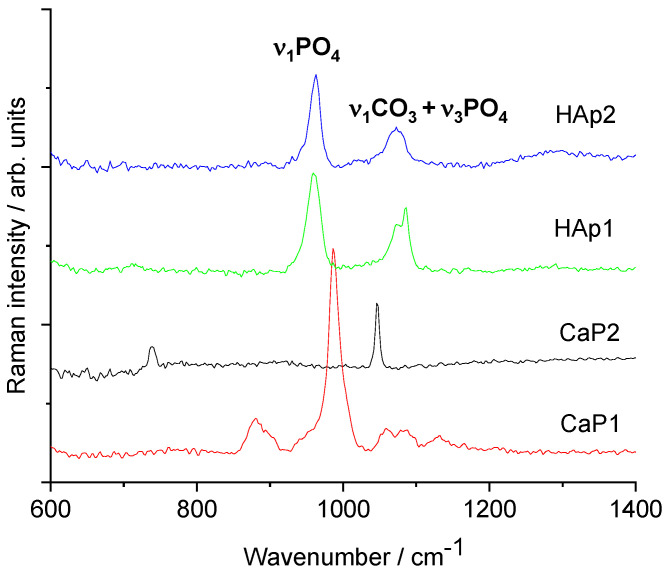
Raman spectra of calcium phosphate powders prepared by different routes.

**Figure 6 nanomaterials-11-03194-f006:**
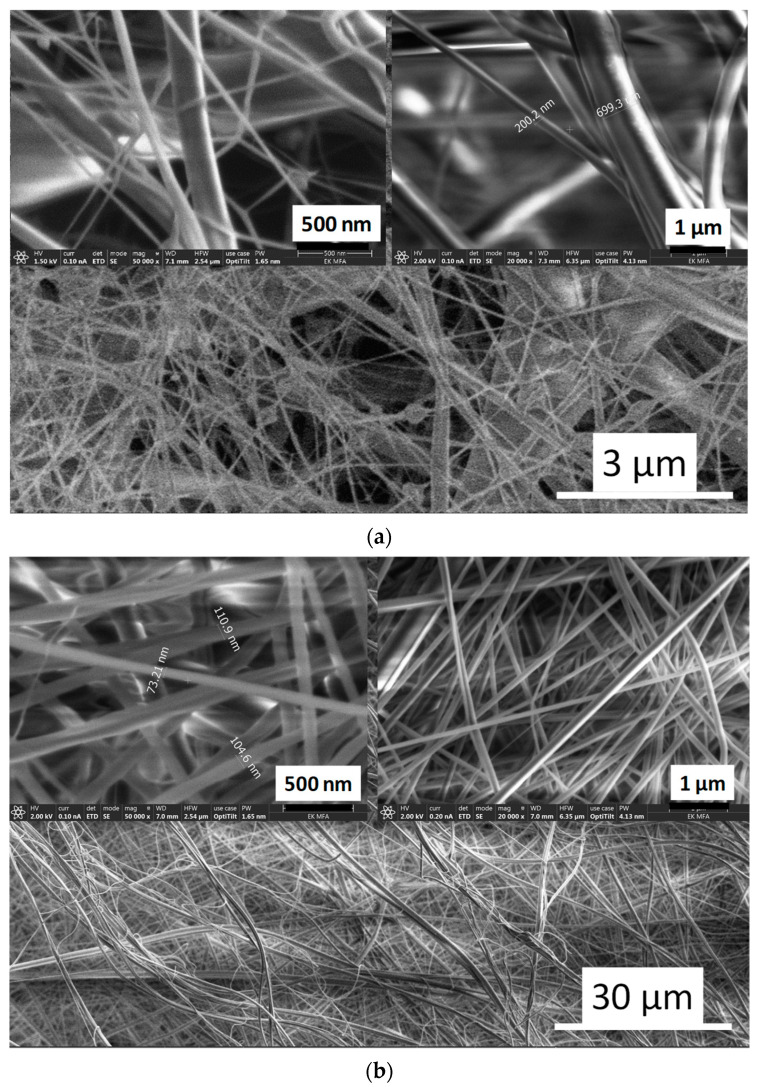
SEM images on pure cellulose acetate (**a**) and PVP (**b**) biopolymer fibres.

**Figure 7 nanomaterials-11-03194-f007:**
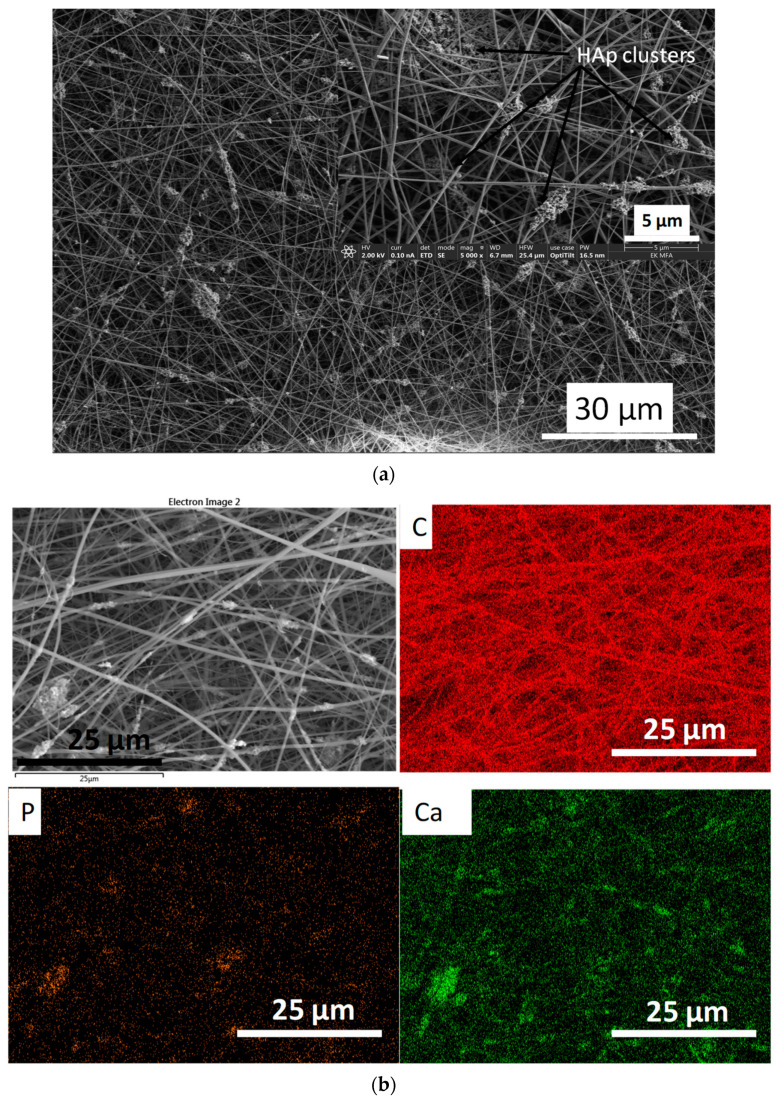
SEM images on cHAp-loaded cellulose acetate (**a**) as well as the corresponding elemental mapping (**b**).

**Figure 8 nanomaterials-11-03194-f008:**
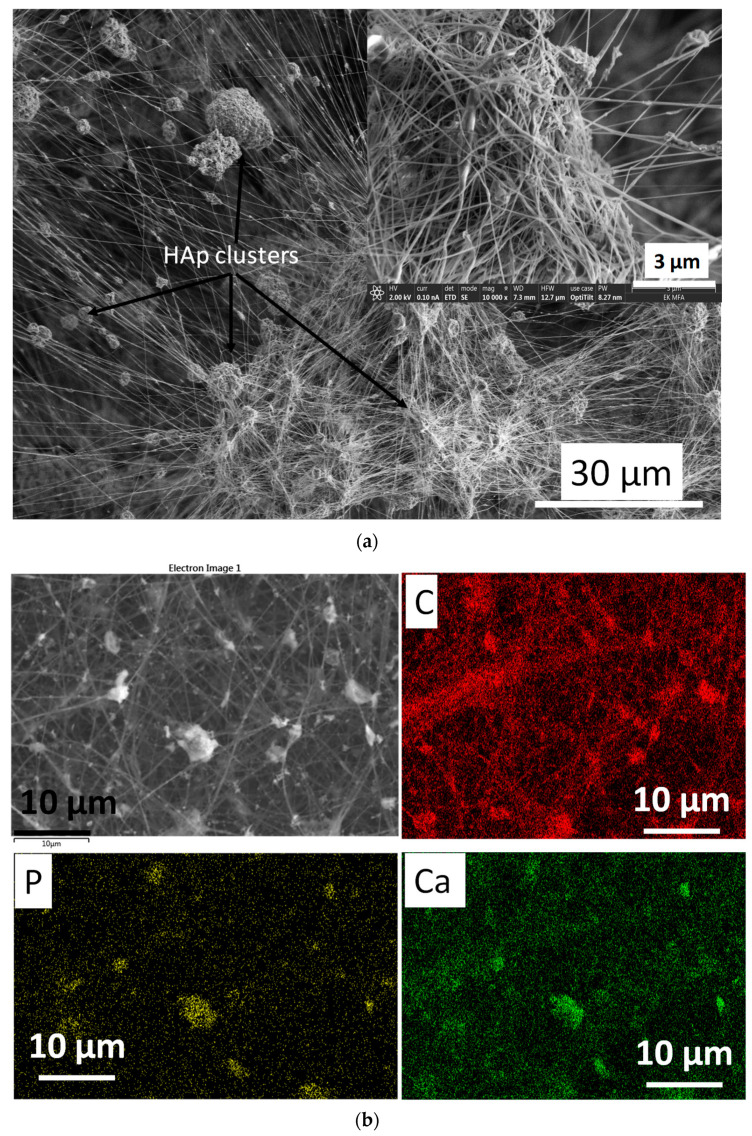
SEM images on cHAp-loaded PVP (**a**) as well as the corresponding elemental mapping (**b**).

**Figure 9 nanomaterials-11-03194-f009:**
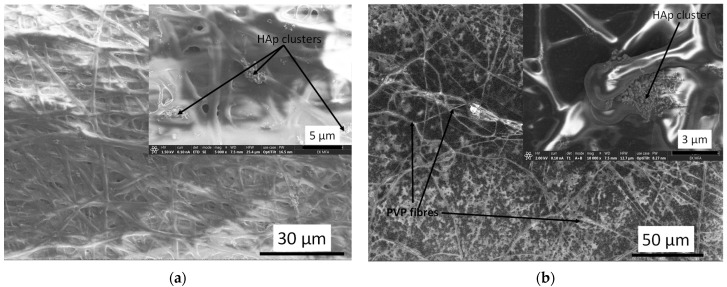
SEM images on cHAp–CA (**a**) and cHAp–PVP (**b**) composites after one-week immersion in PBS solution.

**Table 1 nanomaterials-11-03194-t001:** Elemental composition of CaP samples and the calculated Ca/P elemental ratio.

Elements/ At%	O	C	Ca	P	Total	Ca/P Ratio
CaP1	35.5	4.4	31.7	28.4	100	1.11
CaP2	34.8	1.3	37.8	26.1	100	1.45
HAp1	37.9	3.8	37.9	20.4	100	1.85
HAp2	31.7	5.4	39.8	23.1	100	1.72

## Data Availability

The data used to support the findings of this research study are included within the article.

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
