# Peer review of "Comparison of the Morphological and Structural Characteristic of Bioresorbable and Biocompatible Hydroxyapatite-Loaded Biopolymer Composites"

_nanomaterials, 2021, doi:10.3390/nano11123194_

Round 1

Reviewer 1 Report

Dear authors,

Your paper present very interesting results regarding the physico-chemical

investigation of Calcium phosphate (CaP) based ceramic-biopolymer composites.

In order to improve the quality of your work, I kindly encourage you to

add the following results in the paper:

-Crystallite size-from XRD studies. The results of Rietveld refinement.

-the magnification of SEM images.

-the quantitative and qualitative results of the EDS studies.

-deconvolution of the FTIR spectra in 500-1200 cm-1 spectral region in order to better identify the vibrational bands specific to the CaP structure.

Author Response

We are very grateful and we appreciate the time you devoted to review our manuscript and for the constructive corrections and comments in order to improve our paper.

We have addressed all the comments and improved our manuscript according to the given suggestions.

Comment

Crystallite size-from XRD studies. The results of Rietveld refinement.

Answer:

Thank you for the comment, in our case, Rietveld refinement of the XRD results could not be performed. Instead, the so-called full pattern fitting [D. K. Smith, G. G. Johnson Jr., A. Scheible, A. M. Wims, J. L. Johnson and G. Ullmann, “Quantitative X-Ray Powder Diffraction Method Using the Full Diffraction Pattern,” Powder Diffraction, Vol. 2, No. 2, 1987, pp. 73-77. doi:10.1017/S0885715600012409] of the diffractograms, together with crystallite size calculations were executed using the built-in routine of the Diffrac.EVA program. The manuscript has been completed with the calculated crystallite size values, see in text.

Comment

The magnification of SEM images.

Answer:

Thank you for the suggestion, we added higher magnification SEM images to Fig 2 to be better able to see the morphology of each particles

Comment

The quantitative and qualitative results of the EDS studies.

Answer:

We provided the suggested EDS spectra of all powders and added their evaluation in the text.

Comment

Deconvolution of the FTIR spectra in 500-1200 cm-1 spectral region in order to better identify the vibrational bands specific to the CaP structure.

Answer

We have modified the FT-IR spectra according to the suggested form.

Reviewer 2 Report

The manuscript “Comparison of the morphological and structural characteristic of bioresorbable and biocompatible hydroxyapatite loaded biopolymer composites” is interesting and follows a topical issue of improving the performances of implants by covering them with bioresorbable coatings based on hydroxyapatite-biopolymer combination.

In order to fulfil the requirements for using hydroxyapatite-biopolymer combination (polyvinyl pyrrolidone (PVP) or cellulose acetate (CA)) to promote the osseointegration process the biological evaluation of the obtained structures is mandatory.

Thus, this manuscript could be considered one of the first steps in demonstrating the capabilities of these materials’ combinations in using them for bone tissue engineering.

I marked in the text of manuscript some editing errors for correction.

I recommend publication after minor revision (corrections to minor methodological errors and text editing).

Author Response

We are very grateful and we appreciate the time you devoted to review our manuscript and for the constructive corrections and comments in order to improve our paper.

In order to fulfil the requirements for using hydroxyapatite-biopolymer combination (polyvinyl pyrrolidone (PVP) or cellulose acetate (CA)) to promote the osseointegration process the biological evaluation of the obtained structures is mandatory.

Answer

Yes, of course, we totally agree with you, and we are planning to do biocompatibility tests also on these kind of samples and present them in another manuscript.

Thus, this manuscript could be considered one of the first steps in demonstrating the capabilities of these materials’ combinations in using them for bone tissue engineering.

I marked in the text of manuscript some editing errors for correction.

I recommend publication after minor revision (corrections to minor methodological errors and text editing).

Answer

Many thanks for the corrections, we revised the manuscript as indicated.

Round 2

Reviewer 1 Report

I congratulate the authors for their work and for the outstanding results presented in this paper. My recommendation is: accept for publication in present form.